# *Frankliniella panamensis* (Insecta: Thysanoptera), an Emerging Global Threat or Not? Evidence from the Literature

**DOI:** 10.3390/insects16121230

**Published:** 2025-12-04

**Authors:** Helena Brochero, Megan Gee, Mette-Cecilie Nielsen, David A. J. Teulon

**Affiliations:** 1Facultad de Ciencias Agrarias, Universidad Nacional de Colombia, Bogotá 111321, Colombia; embrochero@unal.edu.co; 2New Zealand Institute for Bioeconomy Science Limited, Private Bag 4704, Christchurch 8140, New Zealand; megan.gee@plantandfood.co.nz (M.G.); mette.nielsen@plantandfood.co.nz (M.-C.N.); 3Better Border Biosecurity (B3), New Zealand

**Keywords:** Thripidae, pest status, quarantine status, pest management, research gaps

## Abstract

*Frankliniella panamensis* is a thrips species of increasing interest as a potential pest of crops in Central and South America and as a contaminant in international trade, especially for ornamentals. We undertake a comprehensive review of information on this species from national, regional, and international sources from both English and Spanish language articles. Despite its clear status as a quarantine pest in some countries, there are many areas of scientific uncertainty about its distribution, biology, ecology, and genetics. There is no consensus as to the pest status of *F. panamensis*. Based on these findings, we identify knowledge gaps and discuss priority areas for future research.

## 1. Introduction

Thrips (Order Thysanoptera), and especially flower-inhabiting thrips (mostly from Family Thripidae), are small insects that include species which are plant pests [1]. Thrips can be invasive [2], readily exploit new habitats [3], and pose substantial challenges in plant trade [4] and pest management [5]. A thrips species of increasing current interest is *Frankliniella panamensis* Hood 1925 owing to (1) a greater understanding of the thrips fauna in Central and South America [6] and (2) its interception in various countries associated with the increasing export of ornamentals from South America and especially Colombia [7]. Confounding this is the close morphological similarity between *F. panamensis* and several other *Frankliniella* species, especially *F. occidentalis* (Pergande) [8], one of the most devastating thrips pests of the last 50 years [9,10], and the combined presence of both *F. panamensis* and *F. occidentalis* in plant exports originating from South America [8,11,12,13].

Many countries and jurisdictions have instigated actions to prevent the establishment of invasive species or to mitigate their impacts if they do establish. Knowledge of the potential risks associated with an invasive species provides the basis for these actions and is often founded on the known impacts of the given species in the geographical area where it is native or recently established [14]. Recent work has emphasized the importance of searching in local language databases in combination with standard searches in international databases [15,16] to gain a comprehensive understanding of invasive species for biosecurity risk assessment, including for thrips [17].

Information on the biology and ecology of *F. panamensis* is scattered in the literature and despite its growing quarantine significance, the pest status of *F. panamensis* is unclear. Therefore, in this study we identify, collate, summarize, and critically analyze information from both national, regional (especially Central and South American countries), and international sources, on the taxonomy, diagnostics, distribution, biology and ecology, pest status, and pest management of *F. panamensis* to support pest risk assessment and to highlight gaps in knowledge for future research.

## 2. Materials and Methods

Our approach involved the combination of a systematic literature review (SLR) or search, with a non-systematic review (non-SLR) or analysis [18]. The main SLR search was carried out using “*Frankliniella panamensis*” AND “*F. panamensis*” OR “*F. panamensis*” (and sometimes each separately) as keywords, with no restrictions other than the search period, which spanned the time from when *F. panamensis* was originally described in 1925 until May 2024. An initial SLR search was carried out in Web of Science (WoS) and Scopus. However, as few articles were found in these search databases, we then carried out the main SLR across databases containing peer-reviewed literature (including WoS, Scopus) licenced to the Universidad Nacional de Colombia (https://unal.edu.co) and The New Zealand Institute for Plant and Food Research Limited (https://www.plantandfood.com; now part of the Bioeconomy Science Institute), as well as freely available (open access) scholarly search tools (e.g., CORE and Semantic Scholar), searches from the internet (using Google and Google Scholar), and other relevant websites. Searches included government websites, university repositories where the species occurs or has been intercepted in horticultural/agriculture exports, and pest regulatory institutions worldwide. A small number of additional articles were found by the authors after the initial searches up until the time of manuscript submission.

All records found for *F. panamensis* from the SLR were read completely, and the reference lists of the relevant literature were examined to identify any important literature from unpublished documents not previously found in the SLR searches. Where appropriate, these were sought and obtained through communication with authors. Google Translate was also used extensively by the non-Spanish speaking authors. For the non-SLR, a critical review and content analysis was conducted on the literature related to *F. panamensis* identified from the SLR and articles grouped according to taxonomy, diagnostics, biology and ecology, pest status, pest management, and border protection. Literature was assigned to such groups following a non-SLR approach [18] based on the concept of exposure, expertise, and experience by the authors and categorized into:Relevant and substantive information: primary source/new data, comprising multiple observations, comprehensive/complete information, direct evidence, species adequately distinguished, and comparative with other literature.Relevant and non-substantive information: secondary source/repeated data, comprising few data observations, limited/partial information, indirect evidence, species not adequately distinguished and not comparative with other literature.Non relevant and non-informative information: for example, documents mentioning *F. panamensis* but not focusing on the species and including literature where *F. panamensis* was mentioned in citation lists—was excluded at this point.

A secondary, smaller search was undertaken for “*Frankliniella stylosa colombiensis*”, a valid synonym of *F. panamensis* in use between 1948 and 1997 (see below), for the entirety of each database until October 2023. In a third search we checked for the digital records of the 87 papers (the number identified at the time of this analysis) that were identified as relevant, but their location in a database was not identified in the initial search. We searched both Google and Google Scholar to determine how accessible they were in these databases. Here we used the keywords or phrases from the title of the manuscript, to identify their location.

No statistical analysis was carried out owing to the limited amount of literature obtained in the search. Based on these findings, we identified knowledge gaps and discussed priority areas for future research. There is a substantial amount of literature on *Frankliniella occidentalis* in Central and South America, e.g., [19,20,21,22], a species that is morphologically very similar to *F. panamensis*, with overlapping hosts and a sympatric distribution. The inclusion of *Frankliniella occidentalis* would have significantly expanded this review, and we were careful to select literature that was primarily focused on *F. panamensis*. This was, however, not always straightforward, as for some subjects (e.g., pest management) the literature may have been equally relevant to both species. A future comparative review of these two species, when more information is available on *F. panamensis*, is likely to be beneficial.

Spanish surnames can be composite, including two surnames, which may or may not be explicitly stated in some publications. Our approach was to cite composite surnames (and their spelling) as they are found in the original publication (or as close as possible) so they can be more accurately identified. We have taken the same approach with diacritics, using the same Spanish accent marks as found (or not) in the original publication.

## 3. Results

**Literature searches.** The initial SLR searches of WoS and Scopus found only five and eight articles, respectively, using the keywords “*Frankliniella panamensis*” AND “*F. panamensis*” OR “*F. panamensis*”. However, the subsequent SLR search using similar keywords across multiple databases found over 800 records. Articles referring to *Frankliniella stylosa colombiensis* from the separate search targeting this taxon were limited to five articles [23,24,25,26,27]. This taxon was not referred to outside the taxonomic literature (see Table 1). From all searches, 89 articles on *F. panamensis* were characterized as being ‘relevant’ for our review (Table 1, Table 2 and Table 3) including articles written in English (*n* = 48) and Spanish (*n* = 37) (sometimes with English summaries/abstracts). About 19 of these included articles contained relevant ‘substantive’ information (Table 1—not including distributional data). Some articles were considered to contain substantive information under one subject (e.g., biology) but not under another subject (e.g., ecology). In general, the more substantive articles on taxonomy and diagnostics were written in English, and the more substantive articles on biology/ecology and pest management were written in Spanish (Table 1). Trade/quarantine-related articles were almost all in English (Table 1 and Table 3). For South America, Colombia was the country with the highest number of publications (*n* = 33), followed by Ecuador (*n* = 4), Panama (*n* = 2), and Costa Rica (*n* = 2) (Table 2). A substantial proportion of articles (13/90) were student dissertations/theses (Table 1), with information from some of these published in scientific journals. These results perhaps reflect the importance of student research in the South American context.

The targeted searches in Google and Google Scholar for the location of the c. 87 ‘relevant’ publications found that only 6 were not found easily in either of these two databases (Figure 1). For publications written in English, these included: two papers in relatively obscure journals [11,25], and one publication from a government regulator [28]. For publications written in Spanish, these included: two student project reports [29,30], and one government newsletter [31]. While many important papers were written in Spanish, references to most of these were found in Google and Google Scholar (i.e., standard international databases) in this subsequent search. However, obtaining full text copies (either printed or digital) was difficult in some instances [32,33,34,35,36].

## 4. Discussion

**Taxonomy.** *Frankliniella panamensis* is placed in the order Thysanoptera, suborder Terebrantia, family Thripidae, subfamily Thripinae [37,38]. Most of the 230 described species from the *Frankliniella* genus are known only from the neotropics, but *F. schultzei* and *F. occidentalis* have been widely distributed around the world [7,9,10].

The taxon *Frankliniella panamensis* was raised in 1925 [39] to describe adult female thrips collected from flowers from Boquete, Panama in February and March 1914. Hood [39] noted that these specimens were close to *F. occidentalis* but distinct enough to require a separate taxon. Mound [40] also noted that *F. panamensis* is one of the nearest relatives of *F. occidentalis*. An additional taxon, *Frankliniella stylosa colombiensis*, was raised by [23], to describe adult females and males collected from a legume in Bogota, Colombia in November 1944. These two taxa were synonymized by [27], with *Frankliniella panamensis* Hood 1925 taking priority. While many authors, including [23,39], provided partial descriptions of *F. panamensis*, the most detailed can be found in [7].

Both *F. panamensis* and *F. stylosa colombiensis* are found in the checklists [24,25,41], but *F. panamensis* is found only in the checklist of thrips from Panama [42] and *F. stylosa colombiensis* is not. Indeed, *F. stylosa colombiensis* does not seem to be used in published material after 1974 [25] even though this taxon was only formally made redundant over 20 years later [27]. The taxon *F. panamensis* was used exclusively for collections/interceptions from Colombia from the 1950s onwards (see Table 3) even though Colombia was the origin of *F. stylosa colombiensis* and Panama was the origin of *F. panamensis*. The lack of the practical use of the taxon *F. stylosa colombiensis* may reflect a general recognition of the unusual placement of a South American subspecies in a North American species taxon (*stylosa*). As an aside, Moulton’s [23] interpretation of *Frankliniella* has resulted in many later synonymies [43,44]. It should also be noted that ref. [23] used the taxa *F. minuta colombiensis* in error (and not consistently) for *F. minuta colombiana* [26].

To date, molecular phylogenies examining taxonomic relationships within the Thysanoptera [45,46,47] have not included *Frankliniella panamensis*, although there are some molecular species comparisons developed for diagnostic purposes that support the distinctiveness of *F. panamensis* from the other *Frankliniella* species examined (see below). The limited collections of *F. panamensis* specimens from different altitudes, geographic locations, and plant hosts have not allowed analysis of the research for the existence of the *F. panamensis* subspecies, as has been found with *F. occidentalis* [48].

**Diagnostics.** *Frankliniella panamensis* is not found in the current online multi-entry keys ‘Pest thrips of the world’ [49] or ‘Pest thrips of North America associated with domestic and imported crops’ [50], although *F. panamensis* was briefly mentioned in the identification guide to ‘Species most likely to be taken on plant material imported into Australia’ in the context of distinguishing it from *F. occidentalis* [51]. *F. panamensis* is, however, found in the LUCID key: Thrips of New Zealand [7], reflecting the biosecurity concern of this species in that country where it has been regularly intercepted in quarantine in imported flowers [8,13].

*F. panamensis* is found in the standard dichotomous keys to the genus *Frankliniella* [23,27] (*F. stylosa colombiensis* is also in [23]) and *F. panamensis* is found in a key to Central and South American thrips species [6]. Other morphological keys have been developed to distinguish thrips species in Colombia [29,52] and Costa Rica [53], and keys have been developed for border interceptions in the USA [54,55]. Further, several publications provided lists of morphological characters that enable the distinction of *F. panamensis* from other species of specific interest, and especially *F. occidentalis* (Table 1).

**Table 1 insects-16-01230-t001:** Literature on *Frankliniella panamensis* ordered by topic. ENG: in English, ESP: in Spanish, OTH: in other language. Bold = substantive text, not bold = non-substantive text (see below for definitions).

Topic	Reference
**Taxonomy** Taxa, synonymies	ENG: **[23,27,39]** **GLO**
Checklists/inventories	ENG: **[24,41]** **NAm selected spp.**, **[25]** **GLO**, [42] PAN all spp., [56] PAN all spp., [37] COL all spp., **[38]** **GLO** ESP: [29]^Ϯ^ & [57] COL regional, [58]^Ϯ^ COL all spp., [59]^Ϯ^ PAN cucurbits, [60]^Ϯ^ COL agroecosystems ---------------------------------------------------------------------------------------------------------------
**Diagnostics** Morphological key	ENG: **[6]** **CAm/SAm**, **[7]** **NZL border**, [23] GLO, **[27]** **(GLO)**, [54] USA border, **[55]** **USA border** ESP: [29]^Ϯ^ & [58]^Ϯ^ COL general, [52] COL greenhouse, [53] CRI avocado, [61] ECU rose
Morphological description	ENG: **[8]** **NZL border**, (7) NZL border, [62] neotropical, [63] RUS border ESP: [29]^Ϯ^ & [58]^Ϯ^ COL general, [53] CRI avocado, [59]^Ϯ^ PAN cucurbit, [61] & [64]^Ϯ^ ECU rose, [65]^Ϯ^ ECU ornamentals OTH: [66] POL border
Molecular characters	ENG: **[8]** **NZL border**, **[13]** **NZL border**, [44] GLO *Frankliniella*, **[67]** **COL avocado** ESP: [68]^Ϯ^ COL border, [69] COL border ----------------------------------------------------------------------------------------------------------------
**Biology/ecology** Life cycle and development	ESP: **[70]** **COL ornamentals**, [71] COL fruit trees
Population dynamics/phenology	ESP: [70] COL ornamentals, **[71]** **COL fruit trees**, [72] COL fruit trees, [73] COL *Chrysanthemum*
Biodiversity (incl. natural systems) (see Appendix A for hosts)	ENG: **[56]** **PAN agroecosystem,** [74]^Ϯ^ COL agroecosystem ESP: [29]^Ϯ^, [57], [58]^Ϯ^ COL moorlandand forest reserve, [60]^Ϯ^ COL cultivated and non-cultivated plants, [75] COL living fences, [76] ECU forest reserve
Other	ENG: [77] COL microbiome ESP: [68]^Ϯ^ COL microbiome
**Pest management** Pest status documented (including lack of pest status)	---------------------------------------------------------------------------------------------------------------- ENG: [20] COL cut flowers ESP: [32] COL wheat and barley; [33] COL wheat, [34] COL oat, wheat and barley, [36] COL floriculture, [70] COL ornamentals, [71] COL *Prunus salicina*, [78] COL wheat and barley, [79]^Ϯ^ COL ornamentals, [80] COL oat, wheat and barley, *Pelargonium*, [81] GTM snow pea, [82] ECU pea, [83] BOL rose, [84] COL rose
Monitoring for IPM (not incl. host plant associations)	ESP: **[71]** **COL plant samples and sticky traps,** **[73]** **COL sticky traps,** **[85]** **COL sticky traps**, [86] GTM plant beating, **[87]****^Ϯ^ COL sticky traps**, [84] COL rose
Other management tactics	**ESP:** **[88]****^Ϯ^ COL cover vs. uncovered** ----------------------------------------------------------------------------------------------------------------
**Border management** Found in pest risk analysis (PRA)	All ENG: [89] refers to PRA for *F. panamensis* for Russia, [90] AUS thrips and orthotospoviruses on fresh fruit, vegetable, cut flowers and foliage imports, [91] AUS cut flowers and foliage, [92] UK limited PRA
Listed as quarantine pest	All ENG: [93] EUR standard, [94] AUS list, [95] RUS list, [96] GLO database, [97] JPN list
Border interceptions in various countries with source indicated	All ENG: USA: [54] from EUR, [98] from COL, [99] from COL, [100] from HND, [101] from COL, [102] from COL and ECU, [103] from COL NLD: [11,12] from COL), UK: [92] no data EUR: [28] from COL NZL: [7] from ?SAm, [8] from COL, [13] from COL, [104] from COL AUS: [7] from ?SAm, [104] from COL JPN: [7] from ?SAm PRI: [105] from COL
Non-compliance notifications	ESP: USA: [106] with regard to ECU ENG: Spain: [107] with regard to COL

Relevant and substantive information: primary source/new data, comprising multiple observations, comprehensive/complete information, direct evidence, species adequately distinguished, and comparative with other literature. Relevant and non-substantive information: secondary source/repeated data, comprising few data observations, limited/partial information, indirect evidence, species not adequately distinguished and not comparative with other literature. Codes for geographical focus of publication: ?—not specified or not clear, GLO—global, CAm—Central America, NAm—North America, SAm—South America, EUR—Europe including none EU countries, AUS—Australia, COL—Colombia, CRI—Costa Rica, ECU—Ecuador, GTM—Guatemala, HND—Honduras, NLD—Netherlands, NZL—New Zealand, PAN—Panama, PRI—Puerto Rico, RUS—Russia, UK—United Kingdom, USA—United States of America, JPN—Japan, POL—Poland, BOL—Bolivia. ^Ϯ^ dissertation/thesis.

Many authors have noted the challenge of distinguishing between *F. panamensis* and *F. occidentalis* because of their similar morphology, their overlapping geographical distributions, and the presence of both species together in border interceptions [6,8]. The distinction of slide-mounted specimens of *F. panamensis* and *F. occidentalis* has been based on various characters with differing degrees of accuracy, but one character has now been found to be consistently different between these two species [8]. For both sexes, the upper surface of the hind coxae bears a small and variable group of microtrichia in *F. panamensis*, but these are not found in *F. occidentalis*. This character is used to distinguish these species in a more recent key by [55], who also provides characters to distinguish *F. panamensis* from other similar South American sympatric species such as *Frankliniella insularis*.

In the past, some authors may have used inadequate characteristics for identification. For example, both *F. occidentalis* and *F. panamensis* have a great deal of intraspecific variation in terms of colour [8,48] and at least one author has indicated that the use of colour has led to considerable confusion in the identification of *Frankliniella* in Colombia [57]. Few voucher specimens appear to have been preserved for *F. panamensis* (Appendix A). In addition, [73] recognized the limitations associated with the morphological identification of specimens captured from sticky traps, an issue that has been recognized for other thrips species [108]. All these issues raise concerns as to the accuracy of morphological identification of *F. panamensis* in some studies prior to 2017.

Several publications describe the use of molecular technologies for *F. panamensis* diagnostics. These include the use of the *Cytochrome c oxidase* subunit I mitochondrial gene (*CO*I) for distinguishing between *F. panamensis* and *F. occidentalis* in border interceptions (adults and immatures) [8], and the *CO*I and the nuclear ribosomal internal transcribed spacer region (ITS) targeting eight thrips species (adults only) in avocado and dandelion in Colombia [67,68]. A short communication summarizes the use of the *CO*I barcode region to distinguish between *F. panamensis* and *F. occidentalis* through a non-destructive specimen protocol (stages not specified) in Colombia [69]. Skarlinsky and Rugman-Jones [44] included *F. panamensis* in a *CO*I sequence analysis to distinguish 23 *Frankliniella* species (stages not specified) and, more recently, ref. [13] generated *CO*I DNA barcode data for 29 thrips species and developed a multiplex real-time PCR assay to distinguish between *F. occidentalis*, *F. panamensis*, *T. tabaci*, and *T. palmi*. The assay from [13] was applicable for single eggs, larvae, and adult samples. Restriction Fragment Length Polymorphism PCR with *CO*I sequences has been developed to differentiate between *F. panamensis*, *F. occidentalis* and *T. tabaci*, allowing for a low-cost screening method between these species (Brochero, unpublished). The limitations of molecular databases for thrips diagnostics have been highlighted by [109], given the limited sequence coverage for the Thysanoptera, and especially Thripidae, to which *Frankliniella* belong. There are few studies that examine the intraspecific diversity of *F. panamensis*, which might provide information for a more accurate diagnosis in relation to other species in the genus *Frankliniella*. Some intra-species variation in *F. panamensis* on avocado plants has been documented by [68,69] based on barcoding gap analysis and ref. [36] speculated on the possibility of the development of hybrids between *F. panamensis* and *F. occidentalis* based on observations of specimens with intermediate morphological characteristics between the two species.

**Country distribution records.** Table 2 lists articles that refer to the presence of *F. panamensis* in various Central and South American countries. *F. panamensis* was first described in Panama [39] although there have been only two more publications with primary collection data from that country, i.e., not including GBIF (Global Biodiversity Information Facility) records [110] (Table 2). *F. panamensis* was first recorded in Colombia by [23] as *F. stylosa colombiensis* and since then there have been numerous publications detailing its presence in Colombia (all as *F. panamensis*) as well as numerous records of border interceptions from Colombia in a range of countries (Table 2). There are also convincing published records of *F. panamensis* from Costa Rica and Ecuador (Table 2). However, the record(s) for Guatemala appear to have been from a few sticky trap specimens (Table 2), which are known to be problematic for accurate thrips identification [73,108]. Records from Honduras are based on one border interception (in the USA), and those from Peru are based on a list on the EPPO (European Plant Protection Organisation) website (no specimen data) [96]. Rogg [83] implied that *F. panamensis* is found in rose crops in Bolivia but with no collection data. We could not find any corroborating information for *F. panamensis* being found in Honduras (e.g., [6,27] or Peru (e.g., [6,27,111,112], Buller pers. comm. 2023). Records of *F. panamenis* from Guatemala, Honduras, Peru, and Bolivia should be confirmed and voucher specimens placed in suitable repositories. *F. panamensis* has not been reported from Brazil [113,114,115] (and there do not appear to be checklists of thrips or *Frankliniella* species from other neighbouring countries such as Nicaragua, Salvador, Belize, Venezuela, Guyana, and Chile. A [110] record of *F. panamensis* from Cameroon is presumably an error and has not been listed in Table 2.

**Table 2 insects-16-01230-t002:** Primary geographical focus of publications on *Frankliniella panamensis*. ENG = in English, ESP = in Spanish.

Country	Primary (Collection Data Provided)	Secondary (No Collection Data)	Border Related
Bolivia		ESP: [83]	
Colombia	ENG: [6,23,67,74,77,110] ESP: [25,29,30,32,33,34,36,52,57,58,60,68,70,71,72,73,75,78,79,80,85,87,88,116,117,118,119,120]	ENG: [25,27,38,96] ESP: [69,84]	ENG: [7,8,11,12,28,98,99,100,101,102,103,104,105,107]
Costa Rica	ENG: [6,110] ESP: [53,121]	ENG: [27,38,96]	
Ecuador	ENG: [110] ESP: [64,65,76,82]	ENG: [27,96]	ENG: [102] ESP: [106]
Honduras			ENG: [100]
Guatemala	ESP: [81,86]		
Panama	ENG: [6,39,42,56,110] ESP: [59]	ENG: [25,27,38,96]	ESP: [31]
Peru		ENG: [96]	

All listed as *F. panamensis* except for [23,25] and [110] (in part), which also include *F. stylosa colombiensis*.

**Altitudinal distribution.** An outstanding feature of *F. panamensis* is its altitudinal preferences. Where altitudinal collection data are recorded, they indicate a strong association of *F. panamensis* for high elevation sites across its geographical distribution. Altitudinal collection data from Colombia, Ecuador, and Panama indicated that *F. panamensis* was mostly found between 1400 and 3600 m (Appendix A). In one study in Panama, it was noted that *F. panamensis* was rarely found at lower altitudes [56]. Similar records for comparable altitudinal distribution of *F. panamensis* (but without specific collection data) are found in [6] for Costa Rica and in [76] for Ecuador. In their extensive surveys for *F. panamenis*, refs. [60] and [119] noted that this species was mostly collected in the very humid pre-montane forest (bmh-PM) (2000–4000 m asl) and pre-montane humid rainforest (bp-PM) (1100–1200 m asl) in the Andean region of Colombia, although actual samples were largely restricted to above 1500 m asl (Appendix A). For five sampling locations in Costa Rica, *F. panamensis* was found only at one site (El Guarco), outside greenhouses at c. 1377 m asl but not at lower sites (all between c. 808–1038 m asl) [121]. Latitude/longitude values reported by [121] were entered into Google Maps (https://earth.google.com/) URL (accessed January 2024) to determine the approximate altitude of these sample sites.

Interestingly, the altitudinal records for *F. panamensis* in Colombia overlapped with the introduced and invasive cosmopolitan thrips pests *F. occidentalis* (2000–3000 m asl), and *T. tabaci* (1500–2500 m asl) [60,119], which may provide some insight into the potential distribution of *F. panamensis* if it invaded other territories. This requires further analysis.

**Plant associations.** Appendix A lists plant species from which *F. panamensis* adults (and some larvae) have been reported. Location, altitude, plant part(s) used for collection, sex and maturity of the thrips, sample date (month and/or year), and sources of the information were included where possible. Associations based on border interceptions were considered separately (Table 3), as these may have been compromised during transit, and associations based on sticky trap capture were excluded, as in these circumstances thrips may be vagrants originating from outside the crop and can be difficult to identify accurately (see above).

**Table 3 insects-16-01230-t003:** Records of border interceptions of *Frankliniella panamensis* including plant genus species from which they were collected; country of origin, country (border) and year where intercepted; and type and number of morphs (A = adults, ♀ = female, ♂ = male, L = larvae, NS = instar or sex not stated). Comments provide information on other thrips species found with *F. panamensis* or other pertinent information.

Plant Genus/Species	Plant Part	Origin	Border	Year	*F. panamensis* Morphs, #	Other Species/Comments	Reference
*Agapanthus* sp.	cut flowers	COL or ECU	USA	1969–1970	NS, >1 of 8		[102]
*Agapanthus* sp.	cut flowers	COL	USA	1971–1972	NS, >1 of 87		[103]
*Althaea* sp.	cuttings	COL or ECU	USA	1969–1970	NS, >1 of 8		[102]
*Antirrhinum* sp.	cut flowers	COL	USA	1971–1972	NS, >1 of 87		[103]
*Alstroemeria* sp.	not specified	COL	NZL	2014–2015	2 A, 4 L	1 L *F. occidentalis*	[8]
*Alstroemeria* sp.	not specified	COL	NZL	?	10 Larvae		[13]
*Aster* sp.	not specified	COL	NZL	2014	1 A	6 L *F. occidentalis*	[8]
*Centaurea cyanus*	not specified	COL	USA	1957–1958	NS, 1		[98]
*Chrysanthemum*	cut flowers	COL or ECU	USA	1969–1970	NS, >1 of 8		[102]
*Chrysanthemum* sp.	cut flowers	COL	USA	1971–1972	NS, >1 of 87		[103]
*Delphinium* sp.	cut flowers	COL	USA	1971–1972	NS, >1 of 87		[103]
*Delphinium* sp.	not specified	ECU	USA	2004–2006	NS	1 notification	[106]
*Dianthus caryophyllus*	cut flowers and branches with foliage	COL	EU	2019	NS, ?	6 separate interceptions	[28]
*Dianthus caryophyllus*	cut flowers	COL	ESP	2019	NS, ?	5 notifications on non-compliance	[107]
*Dianthus* sp.	cut flowers	COL	USA	1971–1972	NS, >1 of 87		[103]
*Dianthus*	not specified	COL	NLD	1987–1993	12 ♀	28 ♀ & 1 ♂ *F. occidentalis*	[11]
*Dianthus*	cut flowers	COL	NLD	1987–1995	NS, 8	Possibly overlap with [11]	[12]
*Eucalyptus* sp.	cut flowers	COL	USA	1971–1972	NS, >1 of 87		[103]
*Gerbera* sp.	not specified	COL	NZL	?	Larva		[13]
*Gladiolus* sp.	not specified	COL	USA	1957–1958	NS, 1	0	[98]
*Gladiolus* sp.	flower	HND	USA	1964–1965	NS, 1	0	[100]
*Gladiolus* sp.	cut flowers	COL	USA	1965–1966	NS, 1		[101]
*Kniphofia* (sic) sp.	cut flowers	COL or ECU	USA	1969–1970	NS, >1 of 8		[102]
*Limonium* sp.	cut flowers	COL	USA	1971–1972	NS, >1 of 87		[103]
*Rosa* sp.	cut flowers	COL or ECU	USA	1969–1970	NS, >1 of 8		[102]
*Rosa* sp.	cut flowers	COL	USA	1971–1972	NS, >1 of 87		[103]
*Rosa* sp.	not specified	COL	NZL	2014–2015	5 A, 5 L	2 A, 2 L *F. occidentalis*	[8]
*Rosa* sp.	not specified	COL	NZL	?	Larva		[13]
*Salvia* sp.	cut flowers	COL	USA	1971–1972	NS, >1 of 87		[103]
*Solidago* sp.	not specified	COL	NZL	2015	1 A	1 L *F. occidentalis*	[8]
*Viola* sp.	not specified	COL	USA	1957–1958	NS, 1		[98]
*Watsonia* sp.	cut flowers	COL	USA	1971–1972	NS, >1 of 87		[103]
*Zantedeschia aeothiopica*	not specified	COL	USA	1961–1962	NS, 1		[99]
*Zantedeschia* sp.	cut flowers	COL	USA	1971–1972	NS, >1 of 87		[103]
Not specified	not specified	EU	USA	1983–1999	NS, 1 or 2		[54]

? indicates data not supplied or not clear. Codes for geographical origin: COL—Colombia, ECU—Ecuador, HND—Honduras. USA—United States, NZL—New Zealand, ESP—Spain, NLD—Netherlands, EU—Europe. Refs. [8,13] only provide qualitative data. In All samples in [13] with sample codes starting with T were from interceptions (Gunawardana pers. comm.). Others without this annotation are not listed here as they were not sourced from border interceptions. Refs. [8,13] appear to have used DNA sourced from the same specimens on some occasions.

The presence of adults alone does not necessarily indicate any sort of host association [122], whereas identification of larval hosts provides a much greater insight into the degree of host plant association [123,124]. Nevertheless, the data suggest that *F. panamensis* exploits many host plants, and the comprehensive list supplied (Appendix A) can form the basis from which a greater understanding of host plant association can be explored, including pollination—as has been suggested by [57]. Adults were often found in flowers (*n* = 78), suggesting the importance of the floral habitat for *F. panamensis*, but this requires further study.

*F. panamensis* adults have been reported from at least 52 families, 95 genera, and 119 plant species (Appendix A), which reinforces the perception [71] that *F. panamensis* is a highly polyphagous species. Plant families well represented (>10 species from each) among the adult host species associations included Asteraceae, Fabaceae, Solanaceae, and Rosaceae. Otherwise, plants were widely spread across other families and included grasses, perennial trees, annual crops, and crops under protective cover or in open fields. The sex of *F. panamensis* in most host plant associations was not differentiated (Appendix A), so little can be concluded about sex ratios under natural conditions. Where the presence of *F. panamensis* larvae was reported, it was not clear how these larvae were identified, as there are no published morphological keys for thrips larvae that include *F. panamensis.* Molecular techniques have provided the evidence of larval plant hosts for *F. panamensis* for a small number of host plant species in traded commodities [8,13] (Table 3).

The vast majority of *F. panamensis* host plant associations refer to cultivated (and mostly introduced) plant hosts, especially those associated with ornamental flower production [29,52,57,58,60,65,116,119,121], medicinal herbs [30,88], fruits [60,71,72,119], and cereals [32,33,34,79,80], indicating a certain degree of host opportunism typical of some polyphagous thrips species [3]. The species has also been found in association with weeds [29,53,57,58,60,71,117]. For example, radish (*Raphanus raphanistrum*) is reported to be an excellent weed host of *F. panamensis* [71]. However, ref. [117] observed more thrips (including *F. panamensis*) on *Chrysanthemum* than weeds.

*F. panamensis* is considered an endemic species of Central and South America and has been recorded from many (c. 52) plant species of Central and South American origin ( Appendix A). For example, *F. panamensis* has been recorded from *Espeletia grandiflora* (Asteraceae), an endemic plant species of moorlands, in forest reserves in Bogotá (Quebrada La Vieja) and Cogua (El Tablón), and several rural areas in the Bogotá savanna [57,58].

Records for border interceptions of *F. panamensis* in the accessible literature listed in Table 3 were much more limited and represented mostly plants from the floriculture trade.

**Life history.** There are few studies detailing the life history of *F. panamensis*. Polyphagous thrips species, such as *F. panamensis*, appear to be polyvoltine and vagile, moving between plants as the climate remains suitable [3]. *F. panamensis* also appears to exhibit the typical thripid life history [125] with egg, two larval, two pupal, and an adult stage, even though this has only been reported in any detail in one study [70]. These authors studied *F. panamensis* in a greenhouse at 24.8 °C and 77.9% RH. Eggs, first instar larva, second instar larva, prepupa, and pupa were reported to last 2.5, 2.5, 5, 2.5, and 3 days, respectively, with the time from egg to adult being about 15.5 days [70]. This developmental time is similar to that for *F. occidentalis* of between 12 and 16 days (depending on diet) at similar temperatures [126,127,128,129,130]. The average longevity of *F. panamensis* adults was 47 days for males and 61 days for females (63 days for females without males, 59 days for females with males), which seems surprisingly long compared with the adult longevity of *F. occidentalis* at similar temperatures [130]. Females without males produced an average of 18 larvae per week and females with males 13.3 larvae per week (copulation was not observed nor were eggs counted) over their lifetime [70]. The reported sex ratio under natural conditions was 4:1 (female/male) [70], reflecting typical arrhenotoky found in thrips [125]. Some of these values were reported again in [71] without clear attribution. Presumably *F. panamensis* is polyvoltine, as there are no reports of quiescence and/or diapause. Two generations per month in summer were noted for *F. panamensis* in eastern Antioquia, Colombia [70]. Specimens have been recorded from all months of the year [58,60,110]. There are surprisingly very few reports of male *F. panamensis* (Appendix A) although ref. [60] reported an overall sex ratio of 2683♀:455♂ (6♀:1♂) in his extensive sampling of *F. panamensis* from a variety of host plants in Colombia. Presumably, reproduction in *F. panamensis* is by arrhenotokous parthenogenesis (unfertilized eggs develop into males), as implied by the statements of [70], but this needs to be confirmed, including for the entire distribution of this species. Reproductive modes in thrips and the cytological mechanisms involved in sex determination are not well understood [131]. For example, the sex ratio for *Thrips tabaci* can vary across the season, latitude, longitude, elevation, and food source [132,133], and it may do so with *F. panamensis* too.

**Abundance.** The size of a population changes in relation to births, deaths, immigration, and emigration, and an understanding of these factors can help inform an effective integrated pest management programme [134]. Host plants, weather, natural enemies, and life-cycle factors can also be important factors contributing to thrips numbers as well as the specific agroecosystem of interest [134]. Few of these factors have been studied in detail for *F. panamensis,* at least as reported in the published literature. The status of the host plants of *F. panamensis* needs particular attention (see above) especially in relation to breeding hosts that will influence population abundance, as does the ability of adults to move between different host plants.

Many thrips pest species are studied in temperate regions, where the four seasons are clearly defined, but the distribution of *F. panamensis* encompasses mountainous areas (including the Andes) with tropical and subtropical latitudes characterized by warm, moist and rainy conditions throughout the year [135]. Factors characteristic of Central and South American biosystems, such as the El Niño and La Niña events [136,137,138], will probably affect the population dynamics and abundance of *F. panamensis*.

Seasonal patterns in abundance. Few studies report on the abundance of *F. panamensis* over time. Both ref. [71] (Cundinamarca, Colombia) and ref. [72] (Boyacá, Colombia) noted that population densities of *F. panamensis* had a very close relationship with the flowering phenology of plum trees, with thrips numbers increasing during flowering. There was no increase over time of *F. panamensis* in a cut-flower crop in La Ceja, Antioquia, Colombia, despite the thrips’ short life cycle, its high reproductive capacity, and the high population that occurs in wild hosts [70], and these authors concluded that the management programme for this thrips species was more or less efficient. In *Chrysanthemum* greenhouses (Antioquia, Colombia) the abundance of the three species, including *F. panamensis*, trapped on sticky boards, showed no pattern over time, plant phenology, or for the varieties evaluated [73].

Thrips invasions into plastic greenhouses have two marked peaks on the Bogotá savanna, Colombia [20,139]: one during February or March and one during August, often coinciding with the end of the dry season and the harvest time for crops planted at the beginning of the rainy season [20]. However, the thrips species composition of these invasions was not stated. Similarly, ref. [120] reported on the number of thrips adults caught over time on sticky traps from four rose greenhouses on the Bogotá savanna, Colombia, but the temporal data they illustrated did not distinguish between the many different species present (up to 12 species per greenhouse and 25 species in total) including two rose greenhouses where *F. panamensis* was reported to be more abundant than *F. occidentalis*.

Abundance in ornamental crops. The abundance of *F. panamensis* in protected ornamental crops and surrounding vegetation deserves specific attention because of the presence of this thrips species in international trade (see below and Table 2 and Table 3). Up to 25 species of thrips have been recorded from ornamental crops in Colombia and Ecuador, including *F. occidentalis*, *T. tabaci*, *F. panamensis*, and *T. palmi* [35,52,61,70,116,117,120] (Brochero unpublished data). Within greenhouse flower crops, *F. occidentalis* has been described as the dominant species on the Bogotá savanna [22,35,52] (Brochero unpublished data) and Antioquia [116]. *F. panamensis* has been observed in high populations in flowers in the open field around the greenhouses [52] but these authors noted that the numbers found in greenhouses were minor. Mound and Isaza [36] commented on the apparent lack of exchange of the thrips species between the inside and outside of greenhouses in the Bogotá savanna and [29] went as far to say that *F. panamensis* was not found inside greenhouses in the Bogotá savanna. A personal communication in [29] quoted Forero, who speculated that *F. panamensis* did not complete its development cycle in greenhouses. In addition, refs. [20,35] reported that *F. panamensis* was the most abundant species outside the greenhouses but provided no supporting data. In contrast, ref. [70] reported that *F. panamensis* was the most abundant species on *Chrysanthemum* in La Ceja, Antioquia, Colombia and refs. [118,120] reported that *F. panamensis* predominated in two out of four rose greenhouses in Bogotá savanna. Godoy et al. [61] recorded several species of thrips present in rose crops in Ecuador but did not indicate the proportion of *F. panamensis* present.

Differences in the proportion of *F. panamensis* and *F. occidentalis* in different studies are possibly explained by different sampling methods. Some researchers [29,35,52,116] appeared to sample flowers directly or from beating flowers, whereas [70,118,120] sampled flying thrips on sticky traps. No study appears to have sampled eggs, larvae, or pupae, which would have provided more precise information on which thrips species were actually inhabiting the crop or which species were merely incidental vagrants. Season, crop type, and management factors may also contribute to differences in the relative species composition inside and outside greenhouses. One study [139] noted that after the plastic cover had been removed from a crop in the Bogotá savanna, the proportion of *F. occidentalis* decreased and the proportion of *F. panamensis* in the crop increased. Calixto Álvarez [57] noted that *F. panamensis* was the most common species in the Bogotá savanna overall, but it was not clear if this statement was made in reference to the inside or outside of the greenhouses. Thrips populations within greenhouses might be influenced by external environmental disturbance of thrips habitats [35]. Another author [61] noted that very high population densities of *F. panamensis* were found in plants located on the perimeter of a rose crop in Ecuador and were more pronounced on the sides close to a contaminated crop. The ability of *F. panamensis* to invade greenhouses may relate to the type of ventilation found in greenhouses in South America (e.g., [140]).

Abundance in non-ornamental crops. Based on beating plants over white trays, *F. panamensis* was reported to be the most common thrips species in covered and uncovered mint crops (*Mentha spicata*) in eastern Antioquia, Colombia (El Retiro, 2175 m asl, Rionegro, 2130 m asl) [88], but there were no measures of immatures to determine if these were breeding hosts. *F. panamensis* was the most abundant species (sampling method not stated) in fruit trees (plum, pear, apple, peach) trees representing 94.2% of the 14 species of thrips recorded [72].

**Border interceptions.** One of the primary drivers for the effective management of *F. panamensis* in Central and South America comes from the quarantine status of this insect. Both Colombia and Ecuador are major exporters of cut flowers (second and third by value, respectively), especially to North America [141], and are subject to quarantine controls in many importing countries [142]. The presence of only a few pests such as thrips (alive or dead), and/or their damage, can cause rejection at ports of entry [19]. Interceptions of insects in floriculture are especially challenging as these large volumes of perishable products are traded around the world, and any delay could compromise on-time deliveries and, therefore, profit. The high risk for quarantine action for infested plants translates into a higher use of insecticides in crop production [20,143].

According to [12], Plant Protection Services showed little interest in Thysanoptera in international trade before 1970, although APHIS interception records show many records for thrips before this time (Table 3). After the spread and establishment of *F. occidentalis* around the world in the 1970s and 1980s [9], the interest of border regulators for thrips increased greatly [12], a period of time that overlapped with the expansion of the South American floriculture industry—the Colombian industry scarcely existed in 1966 [144,145,146].

Table 1 and Table 3 list accessible literature associated with the quarantine status, interception records, and non-compliance notifications of *F. panamensis*. There are likely to be many more if government records were searched to a greater degree (this was not part of our search methodology). For example, as recently as 2024, [55] stated that border interceptions of *F. panamensis* from South America were very common in the US context. Similarly, interception records of *F. panamensis* for the United Kingdom have not been updated since August 2020, and records up to this date have been archived and are not freely available [92]. In all, eight countries, including those from North America, Europe, and Oceania, have highlighted the issue of *F. panamensis* entering their jurisdictions through trade.

The first reported interceptions of *F. panamensis* come from the late 1950s from Colombia [98] long before the expansion of the Colombian floriculture industry. There were no records for *F. panamensis* (or *F. stylosa colombiensis*) in the APHIS List of Intercepted Plant Pests between 1934 and 1958 [147]. From that time, most interceptions of *F. panamensis* were from Colombia, reflecting its dominance in the floriculture trade (Table 3). It is not clear why there are so few interceptions from Ecuador, also a large floriculture-exporting country [142], as the Ecuador floriculture industry is also based in high elevations where *F. panamensis* has been reported in rose crops [61]. Relatively small numbers of *F. panamensis* (compared with *F. occidentalis*) were reported by [65] in export rose crops in Ecuador. Thus, it might be interesting to contrast the growing, harvesting, and postharvest management practices of floriculture crops between Colombia and Ecuador and compare and contrast these to border inspections and interceptions. In one report, *F. panamensis* was intercepted in the USA from Europe [54]—presumably the result of transshipment of plants from South America through the Netherlands.

Except for New Zealand [8,13], we found relatively few accessible raw data on interceptions of *F. panamensis* after 1995, although there were many general references to the presence of this thrips species in the plant trade in the literature over this time (Table 3). *F. panamensis* contributed between 10 and 50% p.a. of the pest interceptions for Australia from Colombian cut flowers and foliage between 2018 and 2021, and 39% (average p.a.) for New Zealand [104]. Hinsley et al. [142] found that data on contaminant border interceptions were not available for many countries even when those countries were approached, partly because of government policies and sensitivities around sharing of these data. The limited data available show that *F. panamensis* was relatively common compared with other thrips species in interception records from Central and South America (Table 3). The current importance of *F. panamensis* as a contaminant in international trade was emphasized by [13], who noted that of the 4000 thrips identifications conducted by the Ministry for Primary Industries (New Zealand) from 2010 to 2021, 1736 were *F. occidentalis*, 565 were *F. panamensis*, 88 were *T. palmi*, and 563 were *T. tabaci*.

Because *F. panamensis* and *F. occidentalis* are so similar and found in the same samples, *F. panamensis* has easily been overlooked, and therefore past interceptions of this species may have been underestimated [11]. Indeed, 1992 Vierbergen [148] and in 1994 Vierbergen [11] both list interceptions of *Frankliniella* from Colombia for overlapping time periods during the 1980s and 1990s, but only ref. [11] lists *F. panamensis* as being present. Similarly, 2003 Nickle [149] did not mention *F. panamensis* as a commonly intercepted species found in the USA but then lists this species in 2004 [54]. Perhaps these inconsistencies represent a sudden recognition of the existence of *F. panamensis*, and the later publications of both authors reflect reanalysis of specimens previously attributed to *F. occidentalis*. Recent molecular protocols that allow the diagnosis of immature and adult forms of *F. panamensis* will help to improve biosecurity risk assessment in the trade of ornamental plants from Central and South America [8,13,44,69].

The pattern of international trade in ornamentals is constantly changing because of customer preferences and regulations in importing countries; the development of new floricultural plants, and the expansion of growing areas in exporting countries; and the emergence of new countries involved in the ornamental trade [142]. Entomological surveillance in ports is a major challenge because of the large volumes involved in the ornamental trade, the short transit times, and also the need for inspectors to inspect food as a primary risk. If there is insufficient information on a species as a potential pest, border surveillance efforts will focus on pests and pathogens they consider to be of high risk. For example, [92] considered *F. panamensis* to be of low risk to the United Kingdom. Similarly, Russia [63,89,95] and Poland [66] considered *F. panamensis* to have a low pest potential for these countries from cut flower imports.

**Pest status.** In general, thrips damage is the result of direct feeding and oviposition on leaves flowers or fruit, transmission of viruses, as well as product contamination in the trade of agricultural and horticultural products [3]. Species within the Thripidae, including many from the *Frankliniella* genus, are effectively pre-adapted as pests by their evolutionary history [3]. They are mainly flower and leaf feeding with a tendency to polyphagy, which provides them with substantial resources to exploit [3]. The ability of these species to quickly colonize and establish large populations makes them particularly important opportunistic and invasive pests [2,3], although the pest status of any particular species is also dependent on geographical area, cultivation practices, and market expectations [150]. Despite the rather limited understanding of the biology of *F. panamensis* (see above) there is every reason to consider this polyphagous ‘flower-thrips’ as an opportunistic and potentially significant pest. However, there is no consensus as to the pest status of *F. panamensis* in either the English or Spanish literature.

*Frankliniella panamensis* is not mentioned in major international texts on thrips pests such as [1], except for a comment by [40] in relation to its being a close relative of *F. occidentalis.* In 1994, [11] stated that damage to plants from *F. panamensis* had not been reported, but its presence in international trade may result in new host plants in new areas with unpredictable results. Later in 2017, [7] reported that *F. panamensis* was not yet recorded as a pest or virus vector, and in 2022 [150] did not list *F. panamensis* as a species that affects human crop productivity. It is not known if *F. panamensis* is a vector of plant viruses [117,151] but since *F. panamensis* has been recorded in protected crops of alstroemeria (Brochero unpublished data) and *Chrysanthemum* [73], both of which have records of plant viruses [151,152], it would seem prudent to assess the virus transmission ability of *F. panamensis*.

*Frankliniella panamensis* is not found in the current online multi-entry keys ‘Pest thrips of the world’ [49] or ‘Pest thrips of North America associated with domestic and imported crops’ [50]. *F. panamensis* was briefly mentioned in an earlier version of ‘Pests thrips of the world’ but only in association with its differentiation from *F. occidentalis* [51]. Whereas *F. panamensis* was listed amongst the *Frankliniella* in The Netherlands (as an interception), it was not included in the associated morphological key [12].

Conversely, *F. panamensis* has been included in generic pest lists for South America [32,33,34,78,80], and there are specific references to it as a pest in several publications. Zenner de Polania [79] reported that *F. panamensis* was of little economic importance and was observed only from time to time in the Bogotá savanna. However, they stated that it could attack flowers that then took on a wilted or burnt appearance and high thrips populations could make it necessary to cut and destroy affected flowers. *F. panamensis* was listed as one of seven thrips species causing damage to snow pea in Guatemala [81], and ref. [52] noted that although *F. occidentalis* was the dominant thrips species in greenhouse flower crops in the Bogotá savanna, *F. panamensis* was included as one of the harmful thrips species that was also present. Conversely, refs. [19,20] did not mention *F. panamensis* as one of the most important pests of Colombian floriculture. *F. panamensis* caused economic loss in high populations to *Chrysanthemum* in La Ceja, Antioquia, Colombia with tissue deformation on *Chrysanthemum* leaves due to the death of cells from feeding nymphs and adults [70]. In Ecuador [82] reported that *F. panamensis* had mild sporadic incidence, was limited to certain regions, and was not subject to mandatory controls on *Pisum sativa*. *F. panamensis* affected the quality of plums (*Prunus salicina*) in Colombia by scraping its surface and sucking the cell content of petals, leaves, and tender fruit, thereby affecting the quality of the fruit [71], and *F. panamensis* was listed as a principal pest of rose cultivars grown in Bolivia but not of *Dianthus* spp., *Chrysanthemum* spp., and *Gladiolus* spp. [83]. While *F. panamensis* was reported to be found in greenhouses of commercial crops, there was no evidence of damage caused by that species [29,58]. Although both *Frankliniella occidentalis* and *F. panamensis* were mentioned as the main pests that affect rose crops in Colombia, no details of their impacts were provided [84]. In summary, in none of the studies listed above did we find any direct (e.g., feeding studies) or indirect (e.g., removal by spray trials) evidence of damage by *F. panamensis* to any cultivated or non-cultivated plant species, so that the pest status of this species is clearly not demonstrated.

**Pest management (including monitoring).** The life-history strategies of opportunist species, such as *F. panamensis*, may place severe constraints on pest management options, and successful thrips control must itself be opportunistic and include a varying range of approaches [3]. However, the development of IPM (integrated pest management) strategies for *F. panamensis* is somewhat challenging and perhaps unnecessary, as the pest status of this insect remains ambiguous for most crops (see above). The difference between *F. panamensis* and any other pest thrips species in terms of plant damage has not been adequately clarified, so research on specific management options for *F. panamensis* may be currently difficult to isolate. This is especially true for *F. occidentalis,* for which thrips pest management actions may be the primary target in many crops but that may equally apply to *F. panamensis* [19,20,22,120,153,154,155].

It is, however, useful to review the specific, although limited, observations and research on *F. panamensis* that might contribute to an IPM system for this species.

Despite the claim of [20] that monitoring and spot treatments brought about a substantial decrease in pesticide use in cut flowers in greenhouses on the Bogotá plateau, the use of insecticides is reported to be the main tactic for control for *F. panamensis* and related thrips species in South America. Thrips management in flower crops for export from the Bogotá savanna is based on regular applications of insecticides [19,22,120] and in deciduous fruit crops in Boyacá [72].

Globally, several thrips species, including *F. occidentalis*, have developed resistance to insecticides as a result of high rates of and frequent application [19,156,157]. Depending on the exposure of *F. panamensis* to the high rates of insecticide applications experienced in some crops in South America, and its ability to move between sprayed and unsprayed habitats, there is a risk that this species may develop insecticide resistance.

Substantially reduced numbers of *F. panamensis* (measured by foliage beating samples) were reported in mint crops (*Mentha spicata*) in Antioquia, Colombia covered with plastic (El Retiro) compared with an open field (Rionegro, 2130 m asl) [88].

Possible biocontrol agents amongst the microbiomes of *F. panamensis* and other thrips species found on avocado in eastern Antioquia, Colombia, were examined as a potential first step towards biological and genetic control strategies [77]. However, the most promising genus, *Wolbachia*, was found in very low relative abundance (0.16%) in *F. panamensis* [77]. A catalogue of Anthocoridae found in the neotropics [158] may provide insight for potential biological control agents for thrips, including *F. panamensis,* but such biological control agents may be disrupted by the intense use of insecticides. Potential mite predators in rose greenhouses and surrounding vegetation in the Bogotá savanna, including in greenhouses where flying adult *F. panamensis* were more abundant than *F. occidentalis,* were documented by [118,120].

Research on plant resistance to thrips in Colombia is limited [153]. The high mobility of thrips and the apparent high degree of polyphagy for *F. panamensis* (see above) suggests that the removal of breeding hosts close to horticultural crops would not be worthwhile, and there remains the need to better identify which species are *F. panamensis* breeding hosts (see above).

Effective colours for trapping *F. panamensis* were found to be white, purple, and yellow (blue not tested) in indoor *Chrysanthemum* by [85], and blue and white (yellow not tested) in outdoor *Prunus sativa* by [71]. Twice as many *F. panamensis* were caught on magenta-coloured traps than blue and yellow traps in indoor *Chrysanthemum* by [73]. Blue sticky traps have been used to monitor *F. panamensis* in floriculture crops in the Bogotá savanna (Brochero unpublished data). Spectral characterization of the adhesive cards was not reported in any of these studies, nor was the type of glue specified, so these results must be interpreted carefully. Coloured sticky traps for thrips monitoring in Central and South America are often non-commercial and self-crafted [22], and as it is known that different glues on different traps are known to affect the relative efficacy of thrips capture [108,159] so such self-crafted traps may have varying degrees of efficacy.

Blue traps with either the thrips aggregation pheromone lure (Thripline™) or with the thrips non-pheromone lure (LUREM TR) were more effective than white traps with similar lures [87]. Traps with lures were marginally better than traps without lures. However, the five thrips species present (including *F. panamensis*) were not differentiated in the analysis.

Sampling of *F. panamensis* was undertaken in outdoor *Prunus sativa* using direct observation and branch shaking over trays [71]. Branch shaking was the least expensive system and had a high correlation of thrips numbers with weed sampling. Direct observation and sticky traps were expensive and had a lower correlation of thrips numbers with weed sampling. Spatial patterns (e.g., aggregated, random) for the different sampling methods were determined over time, but these do not appear to have been applied for sampling protocols [71].

## 5. Conclusions and Future Directions

This is the first comprehensive review of *Frankliniella panamensis*, a Central/South America thrips species as a potential pest and biosecurity concern. Despite its clear status as a quarantine pest in some countries, there are many areas of scientific uncertainty about its distribution, biology, ecology, genetics, and pest status. The following are areas of particular importance for ongoing research.

The literature provides evidence for the current geographical distribution of *F. panamensis* to include Colombia, Costa Rica, Ecuador, and Panama. However, reports of this species in Guatemala, Honduras, Peru, and Bolivia should be supported by more robust data. In addition, surveys in high elevation locations in other Central and South America countries could be undertaken to confirm its presence/absence. Determining current geographical distribution of *F. panamensis* is important to better understand aspects of its ecological requirements, host range, and potential spread.

There is a clear need to establish the feeding and breeding hosts for *F. panamensis*. While adults have been recorded from many plant species from many families, there are few records for immature stages that would inform specific host plant status. Experiments to determine reproductive and feeding status should be undertaken to provide information that would inform on the biology and pest status of this species.

In particular, it is not clear if adult *F. panamensis* found in greenhouses are merely transient, flying in from outside, or are part of an established population within the greenhouse. Careful experiments are needed to examine eggs, larvae, pupae, and adults in protected crops and compare these with adults caught in traps.

Major aspects of the biology of *F. panamensis* are unknown, including temperature/development relationships, thermal thresholds, seasonal phenology, and breeding strategies.

Major aspects of the ecology of *F. panamensis* are unknown, including host plant interactions (e.g., plant phenology, host plant tissue preference), predator/prey–pathogen relationships, response to climate variables (e.g., wind, temperature, humidity), mobility (e.g., within and between plant species and habitats), and host-finding behaviour.

There is no consensus of the pest status of *F. panamensis*. Experiments designed to establish (either direct or indirect) experimental evidence should be undertaken to better understand the pest status of this species, including its ability to transmit plant viruses.

Conversely, the ecological benefits of *F. panamensis* in its area of origin, for example, pollination, need to be understood to ensure that any management tactics have minimal impacts on beneficial ecosystem functions that may accrue from *F. panamensis*.

Only once the pest status of *F. panamensis* for particular crops has been established, can specific management tactics for this species be developed over and above those required for other pest thrips species where these occur together. Clearly, minimizing the infestation of plant parts for export crops will be a major focus.

Despite a range of molecular diagnostics having been developed for all life stages of *F. panamensis*, these have been tested on only relatively few specimens, and especially those from Colombia. The precision of the diagnostics should be verified from specimens from across the broader geographical distribution.

All tools (e.g., diagnostics, monitoring) and tactics (e.g., management programmes) will need to be considered, developed, and implemented in relation to biological (e.g., landscape/crop) and economic (e.g., cost/benefit) parameters for specific locations/countries.

While the taxonomic status of *F. panamensis* at the species level seems to be well established, there may be value in understanding variability at the populations or subspecies level, including over the altitudinal, geographic, and host range of the species, including its hybridisation with *F. occidentalis*.

Given the morphological similarity, sympatric geographical distribution, and apparently similar host plant species of *F. panamensis* with the cosmopolitan pest species *F. occidentalis*, there is an urgent need for all ongoing studies to provide voucher specimens deposited in suitable collections (including in DNA libraries) for any new research.

The invasive success (or not) of *F. occidentalis* can be largely attributable to its superiority in interspecific competition against other thrips species [160]. Given the similar habitats and hosts of *F. panamensis* and *F. occidentalis*, a study examining the competitive interaction between *F. occidentalis* and *F. panamensis* may be particularly informative to help understand biotic and abiotic factors that lead to successful invasion.

From the point of view of broader pest risk assessment, a greater understanding of the biosecurity threat from species inhabiting the ‘tierra fría’, the low-latitude and high-altitude areas of South America with climates similar to many oceanic climates found in temperate areas, would seem to be a worthwhile area for future research.

## Figures and Tables

**Figure 1 insects-16-01230-f001:**
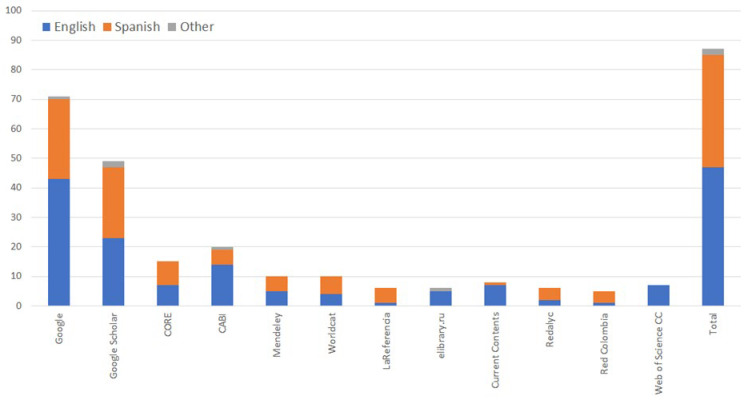
Digital records of ‘relevant’ (substantive and non-substantive) articles on *Frankliniella panamensis* found in digital resources (i.e., databases, catalogues, and repositories).

## Data Availability

Not applicable.

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
