# Peer review of "Frankliniella panamensis* (Insecta: Thysanoptera), an Emerging Global Threat or Not? Evidence from the Literature"

_insects, 2025, doi:10.3390/insects16121230_

Round 1
Reviewer 1 Report
Comments and Suggestions for Authors
Dear Authors,
The review article submitted for peer review is of high scientific value. Based on 161 publications, it highlights gaps in our knowledge of the biology, ecology, and pest status of F. panamensis. A thorough analysis of the available knowledge on this species suggests directions for future research that could prevent its spread in the global ornamental and vegetable trade. Supplementary data contain valuable information on the plants on which F. panamensis has been found. Still, the lack of detailed studies makes it impossible to determine which plants are host plants. Increased attention by cross-border phytosanitary services could prevent the spread of this pest to other areas, as has been the case with species such as F. occidentalis, T. tabaci, T. palmi, T. parvispinus, and others.
I have one technical comment: The species names in lines 383 and 384 should be italicized.
Author Response
Comment #1. I have one technical comment: The species names in lines 383 and 384 should be italicized
Corrected
Reviewer 2 Report
Comments and Suggestions for Authors
This paper is written in a non-concise style (too many unnecessary words and theoretical general information not needed in a paper).
Additionaly it would be better that priority was given to information, instead of who gave the information (the author x said, reported, observed…. instead of placing the name of the author at the end as a citation).
However much work has been done collecting information on this species, sometimes in literature difficult to access. This information may be very useful for those who decide to study the species in the future. The authors should consider the sugestions/ comments listed below:
Line 21- “Despite its clear status as a quarantine pest”- which is the organization/ entity that declares it as a quarantine pest? Where is that written?
Line 22, 23- There is no consensus as to the pest status of F. panamensis.
Line 31- In several line breaks, the syllable division is incorrect (e.g also in lines 117, 125, 129, 256, 354, 372, 393, 411,457, 460, 513, and in many other lines)
Line 39,40 and line 87- South America? South American countries?
Line 71- originated?
Line 159- the authors should consider to call this section “Results and discussion”. Results are not only the statistics on the papers found during the literature search, but also the information (biological, ecological etc) found.
Lines 207- 208- from the Frankliniella genus
Lines 222-224- Sentence not clear
Lines 274-291- It should be clear that this is a footnote (written in smaller print, next to the table)
Line 301- ….(Gunawardana et al. 2017): for both sexes….
Line 351- avoid so many parentheses: from that country, i.e. not including GBIF (Global Biodiversity Information Facility) records (Table…
Line 361- 362- and those from Peru are based on a list of the EPPO
Line 368- panamensis
Line 374- an error
Table 2- unformatted columns
Line 383- All listed as F. panamensis
Line 394- altitudinal
Lines 432-439- It should be clear that this is a footnote (written in smaller print)
Line 454- Replace “insect” by “species”
Lines 498- 503- Consider write in another way, deleting information that may be found in the paper cited:
…study (Zapata-C et al.,1994). These authors studied F. panamensis in a greenhouse at 24.8°C and 77.9% RH: eggs, first instar larva, second instar larva,…
Line 553- Consider changing the subtitle to: Abundance over time
Line 567- plastic greenhouses
Lines 566-576- Information not clearly related to F. panamensis should be deleted
Line 577- Consider changing the subtitle to: Spatial distribution
Line 627- not a good subtitle “Other crops”, in relation to which crops?
633- in fruit trees (plum, pear, apple, peach)
747- The adjective “anecdotal” must be deleted
Lines 771, 775, 777, 782, 786, 789, 792, 795, 797- There is a change in the writing style. Are these integral citations? If so the sentences should be between quotation marks, but it would be desirable to continue with the writting style of the rest of the manuscript.
Line 804- the section that begins here must be rewritten: too much theoretical information, and speculation, and information not related specifically to F. panamensis. If this species reveals to be a pest in the future, than it will be important to explore what we know about IPM of F. occidentalis, to make extrapolations. Right know is premature. However, what it is known about F. panamensis, that may help to control, it should be presented here.
Lines 804-848- use the sentences 811-814, 824-833, 840-842, 844-848. These sentences should be included in the text that comes after line 850.
Line 850- Begin this line: It is useful to review the specific…..
Lines 853-857- This sentence should be deleted
Line 878- Antioquia, Colombia
Line 925- … inform on the biology
Line 940- pest status of F. panamensis.
Line 942- 943- including its ability to transmit
976-980- Sentence not clear
Author Response
Additionaly it would be better that priority was given to information, instead of who gave the information (the author x said, reported, observed…. instead of placing the name of the author at the end as a citation).
Reply - this has largely been changed as a result of incoporating numbered citations
However much work has been done collecting information on this species, sometimes in literature difficult to access. This information may be very useful for those who decide to study the species in the future. The authors should consider the sugestions/ comments listed below:
Reply - thank you we have done so
Line 21- “Despite its clear status as a quarantine pest”- which is the organization/ entity that declares it as a quarantine pest? Where is that written?
Reply - this is found in Table 1 - "listed as a quarantine pest"
Line 22, 23- There is no consensus as to the pest status of F. panamensis.
Reply - change made
Line 31- In several line breaks, the syllable division is incorrect (e.g also in lines 117, 125, 129, 256, 354, 372, 393, 411,457, 460, 513, and in many other lines)
Reply - this is for the type editor to change
Line 39,40 and line 87- South America? South American countries?
Reply - change made
Line 71- originated?
Reply - no change made - 'originating' is correct
Line 159- the authors should consider to call this section “Results and discussion”. Results are not only the statistics on the papers found during the literature search, but also the information (biological, ecological etc) found.
Reply - change made
Lines 207- 208- from the Frankliniella genus
Reply - change made
Lines 222-224- Sentence not clear
Reply - change made to make clearer
Lines 274-291- It should be clear that this is a footnote (written in smaller print, next to the table)
Reply - I have inserted a comment for this comment
Line 301- ….(Gunawardana et al. 2017): for both sexes….
Reply - change made
Line 351- avoid so many parentheses: from that country, i.e. not including GBIF (Global Biodiversity Information Facility) records (Table…
Reply - change made
Line 361- 362- and those from Peru are based on a list of the EPPO
Reply - change made
Line 368- panamensis
Reply - change made
Line 374- an error
Reply - change made
Table 2- unformatted columns
Reply - for type set editor to change
Line 383- All listed as F. panamensis
Reply - change made
Line 394- altitudinal
Reply - change made
Lines 432-439- It should be clear that this is a footnote (written in smaller print)
Reply - I have put in a comment to indicate this - for type set editor to change
Line 454- Replace “insect” by “species”
Reply - change made
Lines 498- 503- Consider write in another way, deleting information that may be found in the paper cited:
…study (Zapata-C et al.,1994). These authors studied F. panamensis in a greenhouse at 24.8°C and 77.9% RH: eggs, first instar larva, second instar larva,…
Reply - change made
Line 553- Consider changing the subtitle to: Abundance over time
Reply - have reworded headings to try and make them clearer but initial heading needs to be general otherwise this would confuse with seasonal changes (a subheading)
Line 567- plastic greenhouses
Reply - change made
Lines 566-576- Information not clearly related to F. panamensis should be deleted
Reply - change made
Line 577- Consider changing the subtitle to: Spatial distribution
reply - changed to Abundance in ornamental crops as better describing the paragraph
Line 627- not a good subtitle “Other crops”, in relation to which crops?
Reply - change made
633- in fruit trees (plum, pear, apple, peach)
Reply - change made
747- The adjective “anecdotal” must be deleted
Reply - change made
Lines 771, 775, 777, 782, 786, 789, 792, 795, 797- There is a change in the writing style. Are these integral citations? If so the sentences should be between quotation marks, but it would be desirable to continue with the writting style of the rest of the manuscript.
Reply - change made
Line 804- the section that begins here must be rewritten: too much theoretical information, and speculation, and information not related specifically to F. panamensis. If this species reveals to be a pest in the future, than it will be important to explore what we know about IPM of F. occidentalis, to make extrapolations. Right know is premature. However, what it is known about F. panamensis, that may help to control, it should be presented here.
Reply - change made - have reduced text in this section
Lines 804-848- use the sentences 811-814, 824-833, 840-842, 844-848. These sentences should be included in the text that comes after line 850.
Reply - done
Line 850- Begin this line: It is useful to review the specific…..
Reply - change made
Lines 853-857- This sentence should be deleted
Reply - change made
Line 878- Antioquia, Colombia
Reply - change made
Line 925- … inform on the biology
Reply - change made
Line 940- pest status of F. panamensis.
Reply - change made
Line 942- 943- including its ability to transmit
Reply - change made
976-980- Sentence not clear
Reply - not sure how to change this!
Reviewer 3 Report
Comments and Suggestions for Authors
The manuscript reviews the English and Spanish literature on Frankliniella panamensis, focusing on its taxonomy, diagnostics, distribution, biology, ecology, pest status, and pest management. The study is relevant and well-constructed, and indeed a huge work. Congratulations.
There are only a few areas where the manuscript should be improved or expanded for clarity.
Simple summary
Line 22-23: ’status F. panamensis’ should be changed to ’status of F. panamensis’
Abstract
Line 40: ’American and especially’ should be changed to ’American countries and especially’
Introduction
In the case of references in brackets, references should be numbered in order of appearance and indicated by a numeral or numerals in square brackets—e.g., [1] or [2,3], or [4–6].
Materials and Methods
This section is correct and well-written.
Discussion
In the case of references in brackets, references should be numbered in order of appearance and indicated by a numeral or numerals in square brackets—e.g., [1] or [2,3], or [4–6].
Lines: 274-281: It is unclear whether this part is related to Table 1 or not. Please clarify.
Lines 282-291: I think that these rows should be either at the end of the Table caption or in the present place, but with smaller letters or closer to the table.
Conclusion
This part is well-written and nicely detailed.
Comments on the Quality of the English Language
The English is fine and does not require any improvement.
Author Response
Simple summary
Line 22-23: ’status F. panamensis’ should be changed to ’status of F. panamensis’
Reply - change made
Abstract
Line 40: ’American and especially’ should be changed to ’American countries and especially’
Reply - change made
Introduction
In the case of references in brackets, references should be numbered in order of appearance and indicated by a numeral or numerals in square brackets—e.g., [1] or [2,3], or [4–6].
Reply - changes made
Materials and Methods
This section is correct and well-written.
Discussion
In the case of references in brackets, references should be numbered in order of appearance and indicated by a numeral or numerals in square brackets—e.g., [1] or [2,3], or [4–6].
Reply - changes made
Lines: 274-281: It is unclear whether this part is related to Table 1 or not. Please clarify.
Reply - have placed a comment there - for type editor to sort out
Lines 282-291: I think that these rows should be either at the end of the Table caption or in the present place, but with smaller letters or closer to the table.
Reply - have placed a comment there - for type editor to sort out